# Sulfane Sulfur Is an Intrinsic Signal for the Organic Peroxide Sensor OhrR of *Pseudomonas aeruginosa*

**DOI:** 10.3390/antiox11091667

**Published:** 2022-08-26

**Authors:** Huangwei Xu, Guanhua Xuan, Huaiwei Liu, Honglei Liu, Yongzhen Xia, Luying Xun

**Affiliations:** 1State Key Laboratory of Microbial Technology, Shandong University, Qingdao 266237, China; 2Food Safety Laboratory, College of Food Science and Engineering, Ocean University of China, Qingdao 266100, China; 3School of Molecular Biosciences, Washington State University, Pullman, WA 99164-7520, USA

**Keywords:** sulfane sulfur, signaling, PaOhrR, DNA binding, MarR family proteins

## Abstract

Sulfane sulfur, including organic persulfide and polysulfide, is a normal cellular component, and its level varies during growth. It is emerging as a signaling molecule in bacteria, regulating the gene regulator MarR in *Escherichia coli*, MexR in *Pseudomonas aeruginosa*, and MgrA of *Staphylococcus aureus*. They are MarR-family regulators and are often repressors for multiple antibiotic resistance genes. Here, we report that another MarR-type regulator OhrR that represses the expression of itself and a thiol peroxidase gene *ohr* in *P. aeruginosa* PAO1 also responded to sulfane sulfur. PaOhrR formed disulfide bonds between three Cys residues within a dimer after polysulfide treatment. The modification reduced its affinity to its cognate DNA binding site. An *Escherichia coli* reporter system, in which mKate was under the repression of OhrR, showed that PaOhrR derepressed its controlled gene when polysulfide was added, whereas the mutant PaOhrR with two Cys residues changed to Ser residues did not respond to polysulfide. The expression of the PaOhrR-repressed mKate was significantly increased when the cells enter the late log phase when cellular sulfane sulfur reached a maximum, but the mKate expression under the control of the PaOhrR-C9SC19S double mutant was not increased. Furthermore, the expression levels of *ohrR* and *ohr* in *P. aeruginosa* PAO1 were significantly increased when cellular sulfane sulfur was high. Thus, PaOhrR senses both exogenous and intrinsic sulfane sulfur to derepress its controlled genes. The finding also suggests that sulfane sulfur may be a common inducer of the MarR-type regulators, which may confer the bacteria to resist certain stresses without being exposed to the stresses.

## 1. Introduction

The signaling role of H_2_S has been extensively studied in mammals, and it is converted to sulfane sulfur that modifies protein Cys thiols to affect the functions of enzymes and gene regulators [1,2]. Sulfane sulfur refers to compounds containing or releasing zero valence sulfur, such as hydrogen polysulfide (H_2_S_n_, n ≥ 2), organic polysulfide (RSS_n_H, RSS_n_R, n ≥ 2), and elemental sulfur [3,4]. It is commonly produced from either sulfide (H_2_S and HS^−^) oxidation [5] or from L-cysteine metabolism [6,7,8]. Cellular sulfane sulfur is maintained in a range, as it readily reduced by cellular thiols, such as glutathione (GSH) to hydrogen sulfide (H_2_S) with the production of glutathione disulfide (GSSG) [9,10]. Excessive sulfane sulfur can also be removed by various enzymes [10,11,12]. Cellular sulfane sulfur in bacteria may reach a maximum higher than 100 μM in the log phase or early stationary phase of the growth [4,13].

Recent reports also support the idea that sulfane sulfur is a signaling molecule in bacteria. Sulfane sulfur has been identified as the effector for gene regulators, such as FisR [14], CstR [15], and SqrR [16], which regulate sulfur metabolism in bacteria. Furthermore, SqrR also regulates gene transfer and biofilm formation in *Rhodobacter capsulatus* [17]. Several gene regulators that regulate other functions also respond to sulfane sulfur. The hydrogen peroxide (H_2_O_2_)-sensor OxyR in *Escherichia coli* also responds to excessive cellular sulfane sulfur to activate the expression of thioredoxin, glutaredoxin, and catalase that remove sulfane sulfur [11]. In *Pseudomonas aeruginosa* PAO1, sulfane sulfur induces MexR to express an efflux pump (MexAB) for antibiotic resistance and enhances the activity of the master quorum sensing activator LasR when the cells enter early stationary phase of the growth [13,18]. In *E. coli*, sulfane sulfur is an intrinsic inducer of the multiple antibiotic resistance regulator MarR [19]. In *Staphylococcus*
*aureus*, MgrA is modified by sulfane sulfur to derepress its controlled genes involved in the virulence and antibiotic resistance [20,21]. The accumulating evidence suggests that sulfane sulfur plays a signaling role in bacteria.

MarR transcriptional regulators are common in bacteria, and they control diverse cellular functions [22,23]. The MarR regulators typically exist as dimers and bind to target palindromic sequences near the promoters to repress the gene expression. MexR, MarR, and MgrA are members of the MarR family. MexR and MarR regulate resistance to multiple antibiotics in *P. aeruginosa* and *E. coli*, respectively [24,25]. MgrA is a global regulator in *S.*
*aureus* [20,21]. OhrR, another MarR family member in *P. aeruginosa*, represses the express of itself and a thiol peroxidase gene *ohr* in the same operon, which is expressed upon exposure to organic hydroperoxides [26]. The thiol peroxidase catalyzes the degradation of organic hydroperoxides with dihydrolipoic acid as the reductant [27]. 

Here, we report that sulfane sulfur is also an inducer of PaOhrR and turns on the expression of PaOhrR-controlled genes when cellular sulfane sulfur is increased. The results suggest that sulfane sulfur is a common signal for the MarR family members. 

## 2. Materials and Methods

### 2.1. Bacterial Strains, Culture Conditions, and Reagents 

Strains and plasmids used in this work are listed in Appendix A. *E. coli* was grown at 37 °C. Kanamycin (50 µg/mL) was added when required. All the primers used in this study are listed in Appendix A. NaHS (H_2_S donor) was purchased from Sigma-Aldrich (Burlington, MA, USA). 

### 2.2. H_2_S_n_ Preparation

H_2_S_n_ was prepared according to a previous report [13]. Briefly, 25.6 mg of elemental sulfur, 32 mg of NaOH, and 44.8 mg of NaHS were added to 20 mL of anoxic distilled water under argon gas. The bottle was sealed and incubated at 37 °C until sulfur was completely dissolved. The H_2_S_n_ concentration was determined by using a cyanolysis method [28]. 

### 2.3. Constructions and Tests of Reporter Systems 

The reporter plasmid pBBR5-OhrR-P*_ohr_*-mKate was constructed with pBBR1MCS5 as the template by placing PaOhrR under a constitutive promoter of the vector and a *mkate* gene after the *ohr* promoter from *P. aeruginosa* in pBBR1MCS5 [26]. The plasmids pBBR5-OhrR-P*_ohr_*-mKate was transformed into *E. coli* BL21(DE3). Plasmid construction was done as reported [13]. The mutant pBBR5-OhrR-C9SC19S-P*_ohr_*-mKate was generated via site-directed mutagenesis as reported [13].

The reporter strains were grown in LB medium at 37 °C with shaking to OD_600nm_ of 2, and 600 µM H_2_S_n_ or NaHS was added. After incubating at 37 °C for additional 2 h, 0.2 mL of the cells was transferred to a 96-well plate and the mKate fluorescence was measured by using the SynergyH1 microplate reader. The excitation wavelength was set at 588 nm and the emission wavelength was set at 633 nm.

### 2.4. Protein Expression and Purification

The ORF of Pa*ohrR* was amplified and subcloned into pET-28a vector between BamHI and XhoI sites. The recombinant protein carried an N-terminal His tag for purification. The cloning was done as reported [13], and site-directed mutagenesis of the cloned Pa*ohrR* was done as reported [13]. 

*E. coli* BL21(DE3) carrying the expression plasmids were cultured in LB medium at 37 °C until OD_600nm_ reached about 0.4–0.6, and then 0.5 mM isopropyl β-D-1-thiogalactopyranoside (IPTG) was added. The temperature was changed to 25 °C for overnight cultivation. Cells were harvested by centrifugation and disrupted through the high-pressure crusher SPCH-18 (Stansted Fluid Power LTD, Harlow, United Kingdom) at 4 °C in ice-cold lysis buffer (50 mM Na_2_HPO_4_, 300 mM NaCl and 20 mM imidazole, pH 8.0). The sample was centrifuged and the supernatant was loaded onto the nickel-nitrilotriacetic acid (Ni-NTA) agarose resin (Invitrogen, Waltham, MA, USA). The target protein was purified following the manufacturer’s instructions. The eluted protein was loaded onto the PD-10 desalting column (GE) for buffer exchange to 20 mM sodium phosphate buffer (pH 7.6). All protein purification was performed under anaerobic conditions and all buffers used were fully degassed. Purity of the proteins was examined via SDS-PAGE.

### 2.5. Electrophoretic Mobility Shift Assay (EMSA)

A 246-bp DNA probe containing the *ohr* promoter sequence was PCR-amplified from the *P. aeruginosa* PAO1 genomic DNA [26]. PaOhrR (10 µM) was treated with 800 µM H_2_S_n_ or 800 µM H_2_S at 25 °C for 20 min. The EMSA reaction mixtures were set up in a final volume of 15 μL of the binding buffer (10 mM Tris, 50 mM KCl, 5% glycerin, pH 8.0) containing different amounts of treated or untreated PaOhrR and 20 nM the DNA probe. After incubating at 25 °C for 30 min, the reaction mixture was loaded onto a 6% native polyacrylamide gel and electrophoresed at 180 V for 1.5 h. The gel was stained with SYBR green I and photographed with a FlourChemQ system (Alpha Innotech, San Leandro, CA, USA). 

### 2.6. Quantification of Cellular Sulfane Sulfur

Cellular sulfane sulfur in *E. coli* at different growth stages was reacted with sulfite to produce thiosulfate that was then quantified according to a reported method [4]. Briefly, samples were mixed with the reaction buffer with sulfite to convert sulfane sulfur to thiosulfate by incubating at 95 °C for 20 min; the buffer without sulfite was used as the control. The produced thiosulfate was derivatized with mBBr and determined by using high-performance liquid chromatography (HPLC) [29].

### 2.7. LC-MS/MS Analysis 

The purification and treatment of PaOhrR used for MS analysis were performed in an anaerobic chamber containing 95% N_2_ and 5% H_2_. The protein was reacted with 10-fold (molar ratio) of H_2_S_n_ at 25 °C for 30 min. The denaturing buffer (0.5 M Tris-HCl, 2.75 mM EDTA, 6 M Guanadine-HCl, pH 8.1) with excess iodoacetamide (IAM) was added to denaturalize protein and block free thiols. The sample was digested by trypsin (Promega, Madison, WI, USA) for 12 h at 37 °C and passed C18 Zip-Tip (Millipore) for desalting before analysis by HPLC-tandem mass spectrometry (LC-MS) by using Prominence nano-LC system (Shimadzu, Kyoto, Kyoto, Japan) and LTQ-OrbitrapVelos Pro CID mass spectrometer (Thermo Fisher Scientific, Waltham, MA, USA). A linear gradient of solvent A (0.1% formic acid in 2% acetonitrile) and solvent B (0.1% formic acid in 98% acetonitrile) from 0% to 100% (solvent B) in 100 min was used for elution. Full-scan MS spectra (from 400 to 1800 *m*/*z*) were detected with a resolution of 60,000 at 400 *m*/*z*. 

### 2.8. Real-Time Quantitative Reverse Transcription PCR (RT-qPCR) 

For RT-qPCR, the cells were harvested at the indicated OD_600__nm_. Total RNA was purified by using a TRIzolTM RNA Purification Kit (12183555, Invitrogen). Total cDNA was synthesized by the HiScript^®^ II Reverse Transcriptase (Vazyme, Nanjing, China). Real-time quantitative reverse transcription-PCR (RT-qPCR) was performed by using a Bestar SybrGreen qPCR Mastermix (DBI Bioscience, Shanghai, China) and LightCycler 480II (Roche, Penzberg, Germany). For calculation of the relative expression levels of tested genes, rplS was used as the reference gene.

### 2.9. Statistical Analysis 

All experiments were performed at least three times in parallel. Data were expressed as means ± standard deviation, and differences between groups were evaluated using Student’s *t* test for individual measurements. Analysis was carried out by using GraphPad Prism v.5 software.

## 3. Results

### 3.1. Sulfane Sulfur Modified PaOhrR and Decreased Its DNA Binding Affinity

The recombinant OhrR from *P. aeruginosa* was overproduced in *E. coli* and purified. EMSA showed that PaOhrR bound to its DNA probe, containing its binding site in a concentration-dependent manner, as previously reported [26], and H_2_S_n_ treatment decreased its affinity to the probe (Figure 1A,B). PaOhrR responded to H_2_S_n_, but not to H_2_S (Figure 1C). A recombinant *E. coli* containing the reporter plasmid pBBR5-OhrR-P*_ohr_*-mKate was constructed. H_2_S_n_ treatment significantly increased the production of mKate fluorescence by the whole cells (Figure 1D), but H_2_S treatment did not (Appendix A). When the PaOhrR was mutated to PaOhrR-C9SC19S, the reporter system completely lost its response to H_2_S_n_ (Figure 1D). 

PaOhrR contains three cysteines: Cys9, Cys19, and Cys121. We generated single and double mutants PaOhrR-C9S, PaOhrR-C19S, PaOhrR-C121S, PaOhrR-C19SC121S, PaOhrR-C9SC19S, and PaOhrR-C9SC121S; PaOhrR and its mutants were purified for EMSA. H_2_S_n_-treatment of the wild-type PaOhrR and the three single mutant proteins C9S, C19S, and C121S reduced their binding to the DNA probe. DTT treatment, which converts disulfide and persulfide back to free Cys thiols, restored their binding to the DNA probe (Figure 2). H_2_S_n_-treatment of PaOhrR-C9SC19S and PaOhrR-C19SC121S did not affect their binding to the DNA probe (Figure 2), indicating that these double mutants no longer respond to H_2_S_n_. The results were consistent with the lack of H_2_S_n_ induction in the *E. coli* strain with pBBR5-OhrR-C9SC19S-P*_ohr_*-mKate (Figure 1D). PaOhrR-C9SC121S did not bind to the DNA probe (Figure 2), suggesting that this double mutant lost its DNA binding ability. The results suggest that the three cysteines may all be involved in sulfane sulfur sensing.

Non-reducing, denaturing SDS-polyacrylamide gel analysis showed that a fraction of PaOhrR formed a cross-linked dimer after H_2_S_n_-treatment (Figure 3A). To identify which Cys residues were involved in forming the dimer, PaOhrR-C9S, PaOhrR-C19S, and PaOhrR-C121S were subjected to non-reducing, denaturing SDS-polyacrylamide gel analysis. Formation of a covalently linked dimer band was visualized on the gel with PaOhrR-C9S and PaOhrR-C19S upon H_2_S_n_ treatment compared with the untreated samples. However, there was no obvious linked dimer bands with PaOhrR-C121S after H_2_S_n_-treatment suggesting that Cys121 is essential for the formation of the interprotomer dimer (Figure 3B). However, the H_2_S_n_-treated PaOhrR-C121S moved slightly faster than the untreated PaOhrR-C121S (Figure 3B), indicating the intraprotomer disulfide bond between Cys9 and Cys19. The dimer structure of OhrR from *Xanthamonas campestris* has been reported [30], and it was used to generate a 3D structure of OhrR. In the homology-based structure of PaOhrR, the sulfur atoms of Cys9 and Cys19 of the same protomer were spatially close at about 16.4 Å, and they were 18.1 Å and 12.8 Å away from the sulfur atom of Cys121 of the second protomer of the protein dimer (Figure 3C). The distance between Cys9 and Cys19 from the two protomers was 36.9 Å (Figure 3D), preventing direct interactions. Thus, an intraprotomer disulfide bond could be formed between Cys9 and Cys19, and interprotomer disulfide bonds could be formed between Cys121 with either Cys9 or Cys19. Cys22 and Cys127 of *X. campestris* OhrR are 15.5 Å apart, and an intraprotomer disulfide bond is formed upon oxidation by organic hydroperoxides, which prevents the oxidized OhrR from DNA binding [30].

To confirm the formation of disulfide bonds, we treated the purified PaOhrR with H_2_S_n_ and analyzed by using LC-MS/MS. A 3+ charged peak (*m*/*z*: 1212.64) corresponding to the Cys^9^-Cys^121^ disulfide-containing peptide (Appendix A), a 4+ charged peak (*m*/*z*: 801.40) corresponding to the Cys^9^-Cys^19^ disulfide-containing peptide (Appendix A), and a 3+ charged peak (*m*/*z*: 1528.12) corresponding to the Cys^19^-Cys^121^ disulfide-containing peptide (Appendix A) were identified. The MS results were consistent with non-reducing, denaturing SDS-polyacrylamide gel analyses that all three cysteine residues were involved in forming disulfide bonds after H_2_S_n_ treatment (Figure 3A,B).

### 3.2. PaOhrR Responded to Intrinsic Sulfane Sulfur

*E. coli* was reported to contain as much as 400 μM sulfane sulfur when grown in LB medium [3,4]. We further detected whether PaOhrR could sense the endogenously produced sulfane sulfur by using the engineered *E. coli* BL21 (pBBR5-OhrR-P*_ohr_*-mKate) and *E. coli* BL21 (pBBR5-OhrR-C9SC19S-P*_ohr_*-mKate). A slight growth delay was found with *E. coli* BL21 (pBBR5-OhrR-C9SC19S-P*_ohr_*-mKate) compared with *E. coli* BL21 (pBBR5-OhrR-P*_ohr_*-mKate) (Figure 4A). The mKate fluorescence was significantly increased in *E. coli* BL21 (pBBR5-OhrR-P*_ohr_*-mKate) when the cells entered the late log phase of growth, but the increase was significantly reduced in *E. coli* BL21 (pBBR5-OhrR-C9SC19S-P*_ohr_*-mKate) (Figure 4B). Given that mKate has a maturation half-time of 75 min [31], the maximal cellular sulfane sufur should be reached at the mid to late log phase of the growth. Indeed, *E. coli* BL21 (pBBR5-OhrR-P*_ohr_*-mKate) contained the highest cellular sulfane sulfur when the cells were in the mid log phase of the growth (Appendix A), similar to the previously results [3,4]. 

The sulfane sulfur level in *P. aeruginosa* PAO1 cells at the early stationary phase (OD_600nm_ of 5.5) is about 4-fold higher than that in cells at the early logarithmic phase OD6_00nm_ of 0.6) [13]. PaOhrR represses the transcription of itself *ohrR* and a thiol peroxidase gene *ohr* in a single opron [26]. The transcription levels of ohrR, ohr, and the next downstream gene *efp*, which encodes elongation factor P [32], in *P. aeruginosa* PAO1 at OD_600nm_ of 0.6 and 5.5 were analyzed by RT-qPCP. The *ohrR* expression in the early stationary-phase cells was 135-fold higher than that in the early logarithmic-phase cells (Figure 5A). The higher *ohrR* should lead to lower *ohr* expression; however, the *ohr* expression in the stationary-phase cells was also much higher (about 4-fold) than that in the logarithmic-phase cells (Figure 5A). No changes in the expression levels of the control gene *efp* either at OD_600nm_ of 0.6 and 5.5 (Figure 5A). The PAO1 cells were also induced with H_2_S_n_. When H_2_S_n_ was added, the *ohrR* and *ohr* expression was obviously increased (Figure 5B). No changes in the expression levels of *efp* after H_2_S_n_ induction (Figure 5B). The results indicate that sulfane sulfur is an intrinsic inducer of PaOhrR, and the regulation of *ohr* by cellular sulfane sulfur is a dynamic process, depending on the level of sulfane sulfur, the ratio of PaOhrR in the modified form, and the PaOhrR concentration.

## 4. Discussion

Cellular sulfane sulfur is an intrinsic signal of PaOhrR of *P. aeruginosa*. In the *E. coli* reporter system, its repression is derepressed after H_2_S_n_ treatment or when the cells enter the mid log phase of growth, but not with the PaOhrR-C9SC19S mutant (Figure 1D and Figure 4). PaOhrR represses the expression of a thiol peroxidase gene. Upon exposure to organic hydroperoxides, the thiol peroxidase is expressed, and it degrades organic peroxides [26]. In *P. aeruginosa* PAO1, the expression of ohrR is also higher at the early stationary phase when the cellular sulfone sulfur is higher than that at the early log phase when the cellular sulfane sulfur level is much lower (Figure 5) [13]. Further, H_2_S_n_ induced the expression of ohrR and ohr in *P. aeruginosa* PAO1 (Figure 5). Our results suggest that the thiol peroxidase is also expressed when cellular sulfane sulfur is high. This response is similar to the repression of MexR for the expression of genes coding for a multiple drug efflux pump for antibiotic resistance in *P. aeruginosa* [13]. The efflux pump is normally expressed after antibiotic challenge [33,34]; however, the expression of the pump also occurs when the cells enter the stationary phase [35]. MexR senses the increased level of cellular sulfane sulfur, which reaches the maximal level at the early stationary phase in *P. aeruginosa*, and allows the expression of the genes coding for the pump [13]. The presence of the pump confers the bacterium to resist antibiotics without induction [35]. Similarly, the PaOhrR controlled genes may also display a growth phase-dependent expression without the induction via exposure to organic peroxides. 

The mechanism of sulfane sulfur sensing by PaOhrR involves the formation of three disulfide bonds (Figure 6). PaOhrR is able to form an intraprotomer disulfide bond (Cys9-Cys19) and two interprotomer disulfide bonds (Cys9-Cys121′ and Cys19-Cys121′) after H_2_S_n_ treatment (Figure 3 and Appendix A). The formation of disulfide bonds reduces or blocks its binding to the target DNA, activating the expression of the repressed genes (Figure 1D). These results are similar to the response of MexR and MarR to sulfane sulfur: *P. aeruginosa* MexR forms an interprotomer disulfide that covalently link the protomers within a dimer [13], and *E. coli* MarR generates disulfide bonds linking two dimers via Cys80 thiols after the treatment with H_2_S_n_ [19]. Besides disulfide bonds, persulfidation has been reported to affect the functions of OxyR and MgrA [11,20].

The above proteins that formed disulfide bonds and persulfides may also respond to other oxidants. Cu^2+^ treatment induced the formation of disulfide bonds between two MarR dimers [36]. Both *P. aeruginosa* OhrR and *X. campestris* OhrR respond to organic hydroperoxides with the formation of disulfide bonds [26,30]. *P. aeruginosa* MexR responds to H_2_O_2_ as well as glutathione disulfide (GSSG) and 2,2′-dithiodipyridine to form interprotomer disulfide bonds between Cys^30^ and Cys^62^ within a MexR dimer [25,34]. *Escherichia coli* OxyR responds to H_2_O_2_ to form OxyR-Cys199-S-OH (cysteine-sulfenic acid) [37], which may further react with Cys208 to form a disulfide bond [38,39], and the oxidized OxyR is active as an activator for the expression of its controlled genes. Therefore, there are similarities between H_2_S_n_ and H_2_O_2_ when they react with protein Cys thiols. H_2_S_n_ reacts with a protein Cys thiol to form the protein Cys persulfide (R-SSH), which reacts with another protein Cys thiol to form a disulfide bond [13]. H_2_O_2_ reacts with a protein Cys thiol to form protein Cys sulfenic acid (R-SOH), which reacts with another protein Cys thiol to form a disulfide bond [38,39]. The similarity predicts overlapping signaling by sulfane sufur and H_2_O_2_. 

Sulfane sulfur also induce the formation of multisulfur links in several gene regulators that do not respond to H_2_O_2_. After reacting with sulfane sulfur, *Cupriavidus pinatubonensis* FisR forms a tetrasulfide crosslinking between Cys53 and Cys64 [14]; *Staphylococcus aureus* CstR generates a mixture of di-, tri- and tetrasulfur links between Cys31 and Cys60 [15]; *Rhodobacter capsulatus* SqrR produces a tetrasulfide link between Cys41 and Cys107 [16]; *P. aeruginosa* LasR creates a pentasulfur link between Cys^201^ and Cys^203^ [18]. The multisulfur links are produced after reacting with sulfane sulfur, as other oxidative reagents, such as H_2_O_2_, organic peroxides, and GSSG, cannot induce the formation of multiple sulfur links between two Cys residues. A multisulfur link suggests the specificity of a sulfane sulfur sensor [40,41].

## 5. Conclusions

Besides sensing organic peroxides, PaOhrR also responds to cellular sulfane sulfur, whose cellular level reaches the maximal level at the early stationary phase of the growth in *P. aeruginosa* PAO1 [13]. The growth phase-dependent expression of the PaOhrR controlled reporter mKate is demonstrated in *E. coli* (Figure 4) and *P. aeruginosa* PAO1 (Figure 5). PaOhrR and several MarR family regulators, such as MarR, MexR, and MgrA are likely sensing both common oxidants and sulfane sulfur (Figure 6). Because cellular sulfane sulfur may reach several hundreds of micromolar at its peak [4,13], it is an important signal for these regulators during normal growth. The sulfane sulfur induced genes may confer the bacteria with resistance to certain stresses without induction. 

## Figures and Tables

**Figure 1 antioxidants-11-01667-f001:**
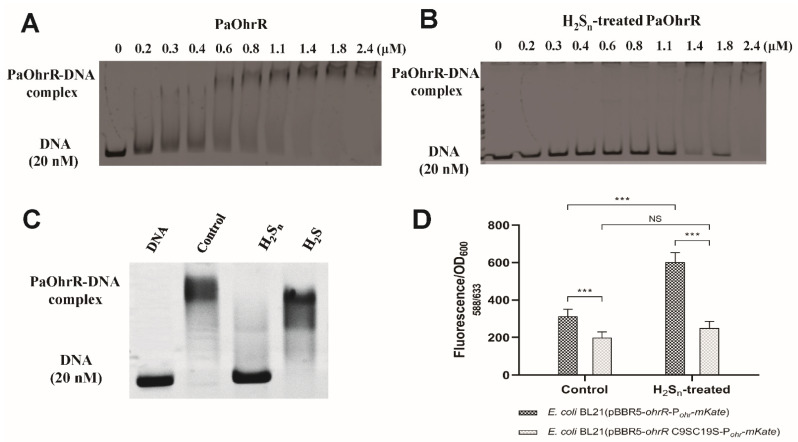
PaOhrR sensed sulfane sulfur in vitro and in vivo. (**A**,**B**) EMSA analysis of the binding of H_2_S_n_-treated and untreated PaOhrR to its DNA probe. Different amount of the treated and untreated PaOhrR was incubated with 20 nM the DNA probe for EMSA. (**C**) H_2_S did not affect PaOhrR binding to its DNA probe. A total of 1.2 µM H_2_S_n_-treated PaOhrR, H_2_S-treated PaOhrR, or untreated PaOhrR (control) was incubated with 20 nM the DNA probe for EMSA. The control was the untreated PaOhrR. (**D**) *E. coli* BL21 containing pBBR5-OhrR-P*_ohr_*-mKate and *E. coli* BL21 containing pBBR5-OhrR (C9SC19S)-P*_ohr_*-mKate were induced at OD_600nm_ of 2 with 600 µM H_2_S_n_ for 2 h. Symbol *** indicates the sample is significantly different from the control (*p* < 0.001). NS = (*p* > 0.05).

**Figure 2 antioxidants-11-01667-f002:**
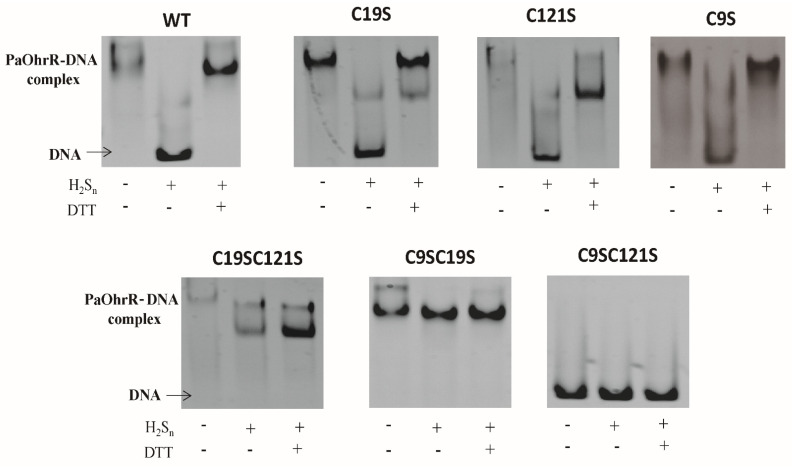
Three PaOhrR Cys residues were involved in sulfane sulfur sensing. The wild type (WT) PaOhrR and its mutants were treated with H_2_S_n_ (800 μM) for 20 min; the proteins (1.2 μM) were then incubated with 20 nM DNA for EMSA. A total of 10 mM DTT was used to reduce H_2_S_n_-modified Cys residues back to free thiols.

**Figure 3 antioxidants-11-01667-f003:**
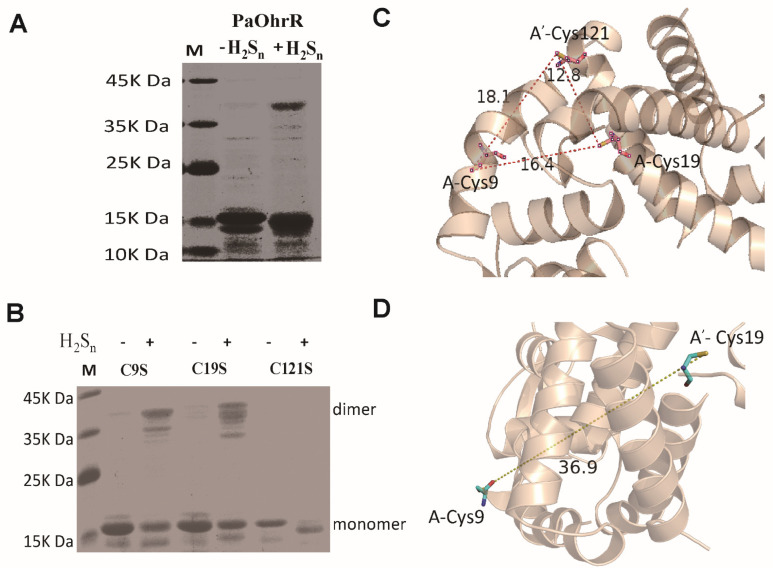
Three cysteine residues of PaOhrR were involved in disulfide formation. (**A**,**B**) Non-reducing, denaturing SDS-PAGE analysis of H_2_S_n_-treated and untreated PaOhrR and its mutants PaOhrR-C9S, PaOhrR-C19S, and PaOhrR-C121S. PaOhrR (25μM) and its mutants were purified and treated with H_2_S_n_ (400 μM) for 20 min and terminated by adding 500 μM iodoacetamide to block free thiols. (**C**,**D**) The 3D structures were modeled by using the SWISS-MODEL method and the *X. campestris* OhrR structure (PDB ID: 2pex) as the template. The distances between the sulfur atoms of the cysteine residues were measured by using PyMOL (v1.5). A and A′ are used to distinguish different monomers in the PaOhrR homodimer.

**Figure 4 antioxidants-11-01667-f004:**
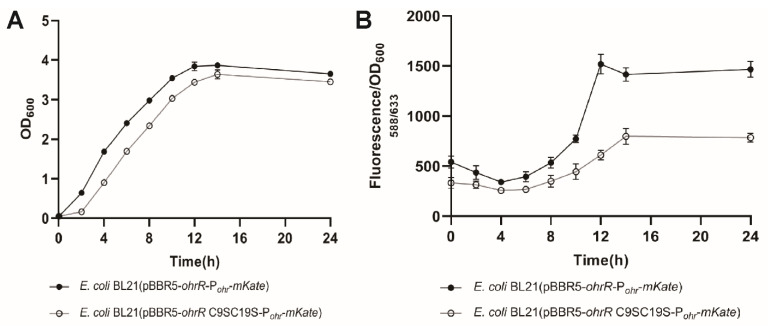
PaOhrR senses endogenous sulfane sulfur. (**A**,**B**) Representative optical density (OD_600nm_) growth curve and mKate fluorescence of *E. coli* BL21 (pBBR5-OhrR-P*_ohr_*-mKate) and *E. coli* BL21 (pBBR5-OhrR(C9SC19S)-P*_ohr_*-mKate suspensions. Cells were grown in LB medium at 37 °C and were analyzed at different growth stages. Data are averages of three experiments with standard deviations.

**Figure 5 antioxidants-11-01667-f005:**
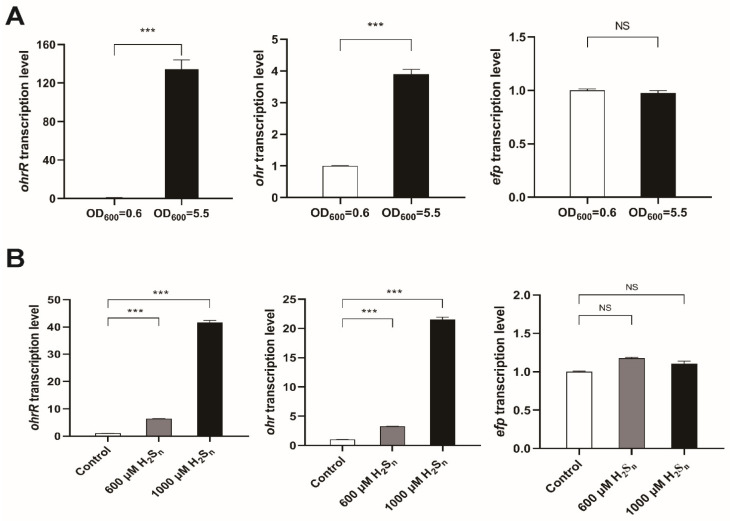
The *ohrR* and *ohr* expression were associated with sulfane sulfur levels in *P. aeruginosa* PAO1. (**A**) RT-qPCR was used to quantify the expression of *ohrR*, *ohr*, and *efp* in PAO1. PAO1 cells were sampled at OD_600nm_ of 0.6 or 5.5 for RNA extraction. (**B**) H_2_S_n_ induced the expression of *ohrR* and *ohr*. The transcripts of *ohrR* and *ohr* were analyzed by RT-qPCR. Cells grew in LB medium until OD_600nm_ of 2 before adding H_2_S_n_ for 20-min induction. The induced and uninduced cells were collected, and RNA was extracted. The *rplS* transcript was used as the internal standard. Unpaired *t* tests were performed (*** *p* < 0.001). NS = (*p* > 0.05). The values are averages of three measures with standard deviations (error bars).

**Figure 6 antioxidants-11-01667-f006:**
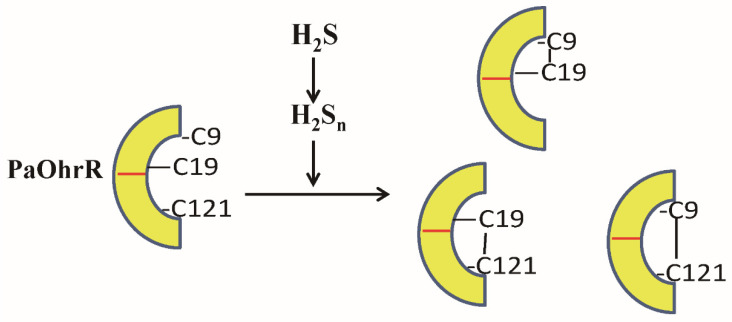
Schematic representations of the PaOhrR dimer that senses sulfane sulfur, forming several disulfide bonds between three key cysteine residues.

## Data Availability

Data are contained within the article or Appendix A.

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
