# Peer review of "Sulfane Sulfur Is an Intrinsic Signal for the Organic Peroxide Sensor OhrR of Pseudomonas aeruginosa"

_antioxidants, 2022, doi:10.3390/antiox11091667_

Round 1
Reviewer 1 Report
The present article entitles “Sulfane sulfur is an intrinsic signal for the organic peroxide sensor OhrR of Pseudomonas aeruginosa”, explain how sulfane sulfur induce OhrR and modulate OhrR-controlled genes. The article is original but can be improved. The introduction can be improved because is a bit chaotic and difficult to understand well the explanation of the MarR family.
The text of the Figure 1 is not correctly described. For example, in the panel C said “H2S did not affect OhrR binding to its DNA probe. The purified OhrR (10 μM) was treated with 800 H2S for 20 min; 1.2 μM untreated, H2Sn-treated OhrR, or H2Sn-treated OhrR was then incubated with 20 nM the DNA probe (60X) for EMSA”. Is very confusing and the control is not explained.
In the Figure 2 the results or the resolution of the results are not clear. For example, in the panel of the mutant C9S DNA band or OhrR-DNA complex band are not clear. Do you have another replicate of the experiment with better resolution or with more clear bands?
On the other hand, how many replicates have you do it per experiment? Is not specify in methods or the figures. In the figure 5 is only specify.
In the figure 5, where is the statistics?? And have to appear in methodology.
Author Response
The present article entitles “Sulfane sulfur is an intrinsic signal for the organic peroxide sensor OhrR of Pseudomonas aeruginosa”, explain how sulfane sulfur induce OhrR and modulate OhrR-controlled genes. The article is original but can be improved. The introduction can be improved because is a bit chaotic and difficult to understand well the explanation of the MarR family.
A: Thank you. We have revised the introduction section.
The text of the Figure 1 is not correctly described. For example, in the panel C said “H2S did not affect OhrR binding to its DNA probe. The purified OhrR (10 μM) was treated with 800 H2S for 20 min; 1.2 μM untreated, H2Sn-treated OhrR, or H2Sn-treated OhrR was then incubated with 20 nM the DNA probe (60X) for EMSA”. Is very confusing and the control is not explained.
A: Thank you for your comments. We moved the protein, H2S, and H2Sn concentrations to the methods section (Line 135-138). The figure legend is simplified.
In the Figure 2 the results or the resolution of the results are not clear. For example, in the panel of the mutant C9S DNA band or OhrR-DNA complex band are not clear. Do you have another replicate of the experiment with better resolution or with more clear bands?
A: Thank you for your comments. We have changed a new picture of the mutant C9S in the manuscript.
On the other hand, how many replicates have you do it per experiment? Is not specify in methods or the figures. In the figure 5 is only specify.
In the figure 5, where is the statistics?? And have to appear in methodology.
A: All experiments were performed at least three times. We have added the statistical analysis in the section of Materials and Methods (Line 171-174).
Reviewer 2 Report
H. Xu et al.
Sulfane sulfur is an intrinsic signal for the organic peroxide sensor OhrR of Pseudomonas aeruginosa.
1. The research presented in this manuscript is focused on investigating whether the organic peroxide sensor OhrR of P. aeruginosa (PaOhrR) also senses the increased level of sulfane sulfur. Using several experimental methodologies, including an E. coli reporter system, EMSA and mass spectrometry, the authors demonstrated that sulfane sulfur controls DNA binding and the repressor function of PaOhrR in vitro and in vivo via disulfide bond formation involving tree cysteines residues. This study provides additional knowledge on the topic of OhrR function in sensing sulfane sulfur in E.coli, published by the authors of this manuscript in Redox Biology in 2019 (The hydrogen peroxide (H2O2)-sensor OxyR also responds to excessive cellular sulfane sulfur to activate the expression of thioredoxin, glutaredoxin, and catalase that remove sulfane sulfur. Hou, N.; Yan, Z.; Fan, K.; Li, H.; Zhao, R.; Xia, Y.; Xun, L.; Liu, H. OxyR senses sulfane sulfur and activates the genes for its removal in Escherichia coli. Redox Biol 2019, 26, 101293).
The main weakness of this study is that the role of OhrR of P. aeruginosa in sensing the level of sulfane sulfur was investigated using recombinant PaOhrR and the E. coli reporter system. The paper would be strengthened by demonstrating endogenous sulfane sulfur production in P. aeruginosa cultured under various growth conditions. Another unanswered question is whether PaOxyR promotes sulfane sulfur reduction in P. aeruginosa (knockout of PaOxyR).
In summary, the study is technically sound, but the experimental set up is inappropriate to address the research question and to reflect the title of the manuscript. Therefore, this manuscript is not appropriate in its current state for publication in Antioxidants.
In addition, there are some minor comments:
1. OhrR of P. aeruginosa should be clearly labelled as PaOhrR thought the text and figures.
2. Include the position of protein markers in all SDS-PAGE gels or WB.
3. More attention should be paid to the preparation of Figures. For example: the images of stained bands of OhrR mutants should be better aligned in Fig. 3B.
Author Response
In summary, the study is technically sound, but the experimental set up is inappropriate to address the research question and to reflect the title of the manuscript. Therefore, this manuscript is not appropriate in its current state for publication in Antioxidants.
A: Thank you for your comments. We checked the transcription levels of two PaOhrR controlled genes by using RT-qPCR and added the data to support our conclusion that sulfane sulfur is an intrinsic induder of PaOhrR in either E. coli or in P. aeruginosa PAO1. The related results were shown in Fig. 5.
In addition, there are some minor comments:
- OhrR of P. aeruginosa should be clearly labelled as PaOhrR thought the text and figures.
A: We have revised it in the manuscript.
- Include the position of protein markers in all SDS-PAGE gels or WB.
A: We have modified Fig. 3 in the manuscript.
- More attention should be paid to the preparation of Figures. For example: the images of stained bands of OhrR mutants should be better aligned in Fig. 3B.
A: Thank you for your comments. We have checked and modified Fig. 3B in the manuscript.
Round 2
Reviewer 1 Report
Authors have accepted all my recommendations. The article is ready to be published.
Reviewer 2 Report
Reasonable cosmetic changes provided and no more is expected after 10 days of revision.
Accept